# Effects of the Washing Time and Washing Solution on the Biocompatibility and Mechanical Properties of 3D Printed Dental Resin Materials

**DOI:** 10.3390/polym13244410

**Published:** 2021-12-16

**Authors:** Na-Kyung Hwangbo, Na-Eun Nam, Jong-Hoon Choi, Jong-Eun Kim

**Affiliations:** 1Department of Orofacial Pain and Oral Medicine, Yonsei University College of Dentistry, Yonsei-ro 50-1, Seodaemun-gu, Seoul 03722, Korea; 1014hwangbo@gmail.com (N.-K.H.); JHCHOIJ@yuhs.ac (J.-H.C.); 2Department of Prosthodontics, Oral Research Science Center, BK21 FOUR Project, Yonsei University College of Dentistry, Yonsei-ro 50-1, Seodaemun-gu, Seoul 03722, Korea; jennynam5703@prostholabs.com

**Keywords:** three-dimensional printing, additive manufacturing, washing solution, washing time, cell viability, cytotoxicity, confocal laser scanning, flexural strength, flexural modulus, scanning electron microscopy

## Abstract

Three-dimensional (3D) printing technology is highly regarded in the field of dentistry. Three-dimensional printed resin restorations must undergo a washing process to remove residual resin on the surface after they have been manufactured. However, the effect of the use of different washing solutions and washing times on the biocompatibility of the resulting resin restorations is unclear. Therefore, we prepared 3D-printed denture teeth and crown and bridge resin, and then washed them with two washing solutions (isopropyl alcohol and tripropylene glycol monomethyl ether) using different time points (3, 5, 10, 15, 30, 60, and 90 min). After this, the cell viability, cytotoxicity, and status of human gingival fibroblasts were evaluated using confocal laser scanning. We also analyzed the flexural strength, flexural modulus, and surface SEM imaging. Increasing the washing time increased the cell viability and decreased the cytotoxicity (*p* < 0.001). Confocal laser scanning showed distinct differences in the morphology and number of fibroblasts. Increasing the washing time did not significantly affect the flexural strength and surface, but the flexural modulus of the 90 min washing group was 1.01 ± 0.21 GPa (mean ± standard deviation), which was lower than that of all the other groups and decreased as the washing time increased. This study confirmed that the washing time affected the biocompatibility and mechanical properties of 3D printed dental resins.

## 1. Introduction

Recently developed computer-aided design/computer-aided manufacturing (CAD/CAM) technology is widely used in dentistry and has made a substantial contribution to the development of dental treatment techniques and restoration quality improvements due to its rapidness and reproducibility [1,2]. Among the processing technologies used for this purpose, 3D printing involves stacking raw materials to produce an object; this technique is also called additive manufacturing. This approach has received attention because it can rapidly produce and prototype various materials that can be 3D printed in a layer-by-layer manner [3]. Additive manufacturing has many advantages, such as wasting less material, emitting less heat and noise, and the absence of wear or the need for the replacement of related consumables, since a burr is not used to remove material, as occurs when a cutting (subtractive) method is used [4,5,6]. In dentistry, additive manufacturing technology is used to manufacture dental models for the implantation of surgical guides, crowns, denture bases, clear aligners, and mouth guards using 3D printed polymers, as well as for the manufacture of metal restorations and dental implants [7,8,9]. Recent advances in ceramic printing technology have also led to the development of ceramic material-based prostheses being considered possible [10].

Among the various 3D printers that utilize additive manufacturing methods, stereolithography (SLA) and digital light projection (DLP) devices produce 3D structures by polymerizing liquid photopolymers with an ultraviolet (UV) laser or a UV-light-emitting diode (LED) light source [6]. SLA is popular thanks to its advantages, such as a high print resolution, smooth surface finish, and precise detail reproduction [6]. SLA printers can accurately draw cross-section patterns on the bottom of a material vat using a numerically controlled microlaser spot, allowing the liquid light-curing resin to solidify in a layer-by-layer manner until the printing process is completed [11]. DLP printers use a laser for liquid polymerization, with a digital mirror device (DMD) controlling the curing laser. The DMD consists of several micromirrors that independently control the moving laser beam. DLP is considered to be a rapid process because it can cure an entire layer in one step using the DMD [12,13]. These SLA and DLP methods require the use of a washing process after printing to remove the residual resin attached to the surface [14]. The manufacturing workflow for vat-polymerization 3D printers consists of three steps: 3D print manufacturing, washing, and post-curing [15]. First, the data are processed into a Standard Tessellation Language (STL) file and used for manufacturing with a layer-by-layer build-up process with a vat-polymerization printer [16]. Next, the support of the printed additive manufacturing polymer is removed and the uncured surface resin is cleaned with a dental appliance using solvents: isopropyl alcohol (IPA) or tripropylene glycol monomethyl ether (TPM) [17]. Finally, a post-curing procedure must be performed to ensure the complete polymerization of the printed object [18].

In the photopolymerization additive manufacturing method, an unpolymerized oxygen-inhibition layer is formed at the interface of the resin exposed to oxygen while the resin is polymerized in air [19]. Additionally, since the object is polymerized using a light source in the vat resin solution, the resin solution remains on the surface of the printed product. Residual dental resin monomers have been reported to exhibit cytotoxicity, causing a continuous inflammatory response and contractions when directly applied to pulp tissue; in addition, they inhibit restorative dentin formation [20,21,22]. They are also known to inhibit heat shock protein 72 expression [23] and induce mitochondrial damage [24] in human monocytes. In addition, uncombined free monomers promote the growth of the dental caries bacteria *Lactobacillus acidophilus*, *Streptococcus sobrinus*, *Streptococcus mutans*, and *Streptococcus salivarius* [25,26]. Local and systemic allergic reactions have been reported in 0.7~2% of patients and dentists [27]. These findings indicate the importance of increasing the biocompatibility of the final product by removing the uncured resin from the surface using a sufficient washing process and reducing residual monomers.

However, excessive washing with alcohol-based washing solutions and resin-dissolving agents may diminish the mechanical properties of the resin. Aging by ethanol is known to significantly decrease the fracture toughness of resins, and aging by a mixture of ethanol and water can also cause the solvent to penetrate the matrix, weakening the resin and impairing other mechanical properties [28]. Furthermore, the denture base polymer—especially those based on self-polymerized polymethyl methacrylate—cracks and dissolves on the surface easily when using solvent and disinfectant ethanol; the interphase region between the PMMA polymer bead and the polymer matrix is reportedly most affected by ethanol [29]. Moreover, immersing the composite resin in ethanol has been shown to lower microhardness and diametral tensile strength, and the immersion solution has been shown to change the physical properties of the resin composite [30].

The washing process for 3D printed dental resin is considered a simple process and its importance has been overlooked. There is currently a dearth of studies that have extensively investigated the biocompatibility and mechanical properties of 3D printed dental resins according to the washing times and washing solutions used. Since 3D printed dental resin must exhibit mechanical properties that can cope with large mastication forces, research is needed to determine the optimal washing solution and washing time for improving biocompatibility without diminishing the mechanical strength. However, this is difficult to achieve consistently because the washing time suggested by each 3D printed resin manufacturer varies, as do the washing equipment and solutions that clinicians have, while the existing recommendations are also insufficient.

The purpose of the present study was to determine the effects of the washing solution (IPA and TPM) and time used on the mechanical properties and biocompatibility of photocurable resins that can be used to 3D print denture teeth or a crown and bridge. The null hypothesis was that the mechanical properties and biocompatibility of 3D printed photopolymerizable dental resin would not vary significantly with the type of washing solution and washing time used.

## 2. Materials and Methods

### 2.1. 3D Specimen CAD Design and Printing

The 3D printed specimen file used in this study was designed using the CAD software (Meshmixer, Autodesk, San Rafael, CA, USA) before printing. The disks used in the biocompatibility tests and scanning electron microscopy (SEM) surface analyses were designed to have a diameter of 9 mm and a thickness of 2 mm. For the mechanical property test, a bar with a length of 25 mm, a width of 2 mm, and a thickness of 2 mm was designed according to the ISO 10477 standard, and the designed specimen was saved as an STL file (Figure 1). The NextDent C&B (ND) resin specimen was printed using a dedicated DLP 3D printer (NextDent ND5100, Vertex-Dental, Soesterberg, the Netherlands), with a 405 nm UV LED used as the light source. An SLA 3D printer (Form 3, Formlabs, Somerville, MA, USA) was used to print the specimen with Formlabs Denture Teeth A2 (FL) resin, and a 405 nm UV LED light with a 250 µW laser power were used as the light source. All specimens were 3D printed at a thickness of 50 μm (Table 1).

### 2.2. Washing the 3D Printed Specimens and the Post-Curing Process

Before washing, the support structures and residual resin of each 3D printed specimen were removed; next, it was placed in a 3D printing washer (Twin Tornado, MEDIFIVE, Incheon, Korea) and washed with an IPA or TPM solution for 3, 5, 10, 15, 30, 60, or 90 min to remove the residual resin monomer from the surface (Table 2). A nonwashing group was used as a control group (CTR). After washing with 90% IPA, alcohol was evaporated in air, and after washing with the TPM solution, the sample was rinsed with water or lightly brushed according to the manufacturer’s recommendations. The specimen was then post-cured for 30 min at 60 °C in a UV light polymerization chamber (FormCure, Formlabs) using a 405 nm light source (13 multidirectional LEDs) (Figure 2).

### 2.3. Biocompatibility Test

#### 2.3.1. In Vitro Cell Culture and Cell Line

Cell viability and cytotoxicity assays were performed to evaluate the biocompatibility of the molecules released from the 3D printed specimens according to the washing time. The 3D printed specimens were manufactured according to the manufacturer’s instructions for the resin suggested in ISO 7405 and research by Ian Gibson, I.G., and manufactured with a diameter of 9 mm and a thickness of 2 mm to fit the size of a 48-well plate. Primary human gingival fibroblasts (HGFs; PCS-201-018, ATCC, Manassas, VA, USA) were used with Dulbecco’s modified Eagle medium (WelGene, Daegu, Korea) containing 10% fetal bovine serum (Thermo Scientific, Waltham, MA, USA), 5% penicillin/streptomycin (100×, WelGene), and 5% MEM nonessential amino acid solution (100×, WelGene). The HGFs were incubated at 37 °C in a 5% CO_2_ and 95% air atmosphere and at 100% relative humidity. The cell culture medium was changed every 2 or 3 days, and when the cell confluency reached 85–90%, the culture was treated with 0.25% trypsin-ethylenediaminetetraacetic acid solution (Trypsin-EDTA, 1×, WelGene). The separated cells were centrifuged to remove the old medium and trypsin-EDTA solution and then diluted to a density of 5 × 10^4^ cells/mL in fresh medium. Five specimens were used in each group, and the diluted cells were seeded on 48-well plates containing sterile, circular 3D printed specimens. The plates were incubated for 24, 48, and 72 h at 37 °C and 5% CO_2_ before analyzing cell viability (*n* = 15 per group), and cytotoxicity (*n* = 15 per group).

#### 2.3.2. Cell Viability and Cytotoxicity Assays

The cell viability was measured using the CELLOMAX™ viability assay kit, which is based on the tetrazolium salt (2-(2-methoxy-4-nitrophenyl)-3-(4-nitrophenyl)-5-(2,4-disulfophenyl)-2H-tetrazolium) and monosodium salt (WST-8) (Precaregene, Hanam, Kyungido, Korea). After 24, 48, and 72 h of incubation, 50 μL of CELLOMAX™ solution (CELLOMAX™ viability assay kit, Precaregene) was added to the 48-well plate with the specimen. Wells in which only cells without specimens were cultured were used as a positive control group, and incubated at 37 °C for 90 min according to the manufacturer’s instructions. Next, 100 μL of the medium from each well was transferred to 96-well plates and the absorbance was measured at 450 nm using a microplate reader (VERSA max, Molecular Devices, Sunnyvale, CA, USA). The percentage of cell viability was calculated as follows:$$\mathrm{Cell}\;\mathrm{viability}\;\left(\%\right)=\left(\frac{{\mathrm{OD}}_{\mathrm{test}\;\mathrm{sample}}-{\mathrm{OD}}_{\mathrm{blank}}}{{\mathrm{OD}}_{\mathrm{control}}-{\mathrm{OD}}_{\mathrm{blank}}}\right)\times 100$$

The cytotoxicity of the 3D printed resins to HGFs was evaluated using a lactate dehydrogenase (LDH) release assay kit (Quanti-LDH™ Cytotoxicity Assay Kit, BIOMAX, Seoul, Korea). From 48-well plates cultured for 24, 48, and 72 h, 20 μL of supernatant culture medium was transferred to new 96-well plates. In a high control group, cells were lysed according to the manufacturer’s instructions, and in a low control group, only cells were cultured in wells without specimens. After this, 100 μL of the LDH substrate mixture was added and reacted at room temperature for 30 min, and the absorbance of the resultant solution was measured at 450 nm. The percentage of cytotoxicity was calculated as follows:$$\mathrm{Cell}\;\mathrm{Cytotoxicity}\;\left(\%\right)=\;\left[\frac{\left({\mathrm{OD}}_{\mathrm{cells}\;\mathrm{with}\;3D\;\mathrm{printed}\;\mathrm{resin}}-{\;\mathrm{OD}}_{\mathrm{background}\;\mathrm{control}}\right)-({\mathrm{OD}}_{\mathrm{low}\;\mathrm{control}}-{\;\mathrm{OD}}_{\mathrm{background}\;\mathrm{control}}{}_{\;})}{\left({\mathrm{OD}}_{\mathrm{high}\;\mathrm{control}}-{\mathrm{OD}}_{\mathrm{background}\;\mathrm{control}}\right)-({\mathrm{OD}}_{\mathrm{low}\;\mathrm{control}}-{\mathrm{OD}}_{\mathrm{background}\;\mathrm{control}})}\right]\times 100$$

#### 2.3.3. Confocal Laser Scanning Microscopy Analysis

HGFs were incubated for 24 h in a dish containing sterilized specimens, washed three times with PBS, and fixed with 4% paraformaldehyde in PBS for 15 min at room temperature. Next, the sample was washed three times for 5 min each, treated with 0.1% Triton™ X-100 in PBS for 15 min, and then washed three times with PBS again. For cytoskeletal quantification, phalloidin (Alexa Flour 488^®^ Phalloidin, Invitrogen, Thermo Scientific) was used to fluorescently stain the cytoskeleton via its binding to F-actin according to the manufacturer’s instructions. DAPI (4′,6-diamidino-2-phenylindole), which binds to the AT regions of DNA and emits a blue fluorescence, was then applied for nuclear counterstaining. Finally, the specimens were treated with antifade mounting media (VECTASHIELD, Vector Laboratories, Burlingame, CA, USA) and imaged using confocal laser scanning microscopy (CLSM; LSM 700, Carl Zeiss Microscopy, Jena, Germany).

### 2.4. Flexural Strength and Flexural Modulus Test

For the three-point flexural bend test, 15 bar specimens per group (*n* = 15) were stored at 37 °C for 24 h before performing the test according to the ISO 10477 standard. A universal tester (Model 3366, Instron Corporation, Norwood, MA, USA) equipped with a 10 kN load cell was used at a crosshead speed of 1 mm/min. The specimens were placed on two round supports parallel to each other with a 20 mm separation. A uniaxial load was applied to the center of the specimen until it broke, then the maximum force before fracture was recorded in Newtons. The flexural strength (*σ*), in megapascals, and the flexural modulus (*E*), in gigapascals, were calculated as follows [31]:$$\sigma =\frac{3FL}{2wh^{2}}$$
$$E=\frac{FL^{3}}{4wh^{3}d}$$
where *F* is the load at a selected point of the elastic region of the stress–strain plot in newtons, *L* is the span length between the supports in millimeters, *w* is the width of the specimen in millimeters, *h* is the height of the specimen in millimeters, and *d* is the deflection of the specimen in millimeters (Figure 3).

### 2.5. *Scanning Electron Microscopy* Analysis

After each washing, the surface of the post-cured specimen was analyzed using SEM (JEOL-7800F, JEOL, Tokyo, Japan) as part of a qualitative SEM image analysis. The specimen was mounted after applying a carbon adhesive and coating it with gold palladium. Representative images were obtained at 40× and 1000× magnifications in each group.

### 2.6. Statistical Analysis

The normality of all the data was confirmed by performing a Shapiro–Wilk test. Data from the cell viability (*n* = 480) and cytotoxicity (*n* = 480) assays were analyzed by a three-way ANOVA using standard statistical software (version 25.0, SPSS Statistics, IBM, Armonk, NY, USA), followed by the Bonferroni and Scheffe tests. Through this, the interaction between washing time, washing solution, and materials and the effect of variables on biocompatibility were confirmed. A one-way ANOVA was also used to confirm differences in the experimental results according to the washing time within each group (α < 0.05). In order to confirm the effect of washing time, washing solution, and material on mechanical properties, data on flexural strength (*n* = 480) and flexural modulus (*n* = 480) were analyzed using a three-way ANOVA, followed by the Bonferroni test and Tukey multiple-comparison test. In addition, a one-way ANOVA was used to confirm the differences according to the washing time within various groups.

## 3. Results

### 3.1. Biocompatibility Test

#### 3.1.1. Cell Viability Assay

HGFs were cultured on the surface of the 3D printed specimens for 24, 48, and 72 h, and the cell viability was found to vary with the washing time and the 3D printed resin used (Figure 4). A three-way ANOVA revealed that the cell viability varied significantly with the washing time (F = 216.669, *p* < 0.001) and material used (F = 79.899, *p* < 0.001), but not with the washing solution used (F = 1.298, *p* = 0.255) (Figure 4B). There were significant interactions between the material and washing solution (F = 164.055, *p* < 0.001), material and washing time (F = 6.373, *p* < 0.001), and washing solution and washing time (F = 3.201, *p* = 0.003). There was also a significant interaction effect between the washing time, resin, and washing solution (F = 12.311, *p* < 0.001). Cell viability varied significantly with the washing time (Figure 4A). The cell viability was lowest in the control group at 24.38 ± 6.02%, then increased significantly with the washing time. The cell viability was significantly higher in the ND group (47.11 ± 15.76%) than in the FL group (41.60 ± 19.58%) (Figure 4C).

Figure 5 shows the results of a one-way ANOVA that was used for verifying the differences in cell viability according to the washing time used for each material and the washing solution. The analysis revealed that cell viability increased with the washing time in all groups. In the ND-IPA group, the cell viability was lowest in the control (23.69 ± 2.77%) and was significantly higher for washing times of 30 min (61.86 ± 9.36%), 60 min (69.14 ± 11.79%), and 90 min (74.75 ± 12.31%) (Figure 5A). In the ND-TPM group, cell viability was significantly lower for the washing times of the control (29.47 ± 2.16%) and 3 min groups (30.38 ± 2.16%), and significantly higher for the washing times of the 60 min (53.77 ± 7.55%) and 90 min groups (63.81 ± 12.01%) (Figure 5B). In the FL-IPA group, the cell viability was significantly lower for the washing times of the control (23.73 ± 6.47%), 3 min (24.12 ± 6.61%), and 5 min groups (25.44 ± 7.12%), and significantly higher for the groups with a washing time of 90 min (61.15 ± 8.22%) (Figure 5C). In the FL-TPM washing group, the cell viability was significantly lower for the washing times of the control (20.64 ± 7.43%) and 3 min groups (22.77 ± 7.09%), and significantly higher for the washing times of the 60 min (66.10 ± 8.36%) and 90 min groups (80.79 ± 15.43) (Figure 5D). The data for cell viability after culturing for 24–72 h are listed in Table 3.

#### 3.1.2. Cytotoxicity Assay

The cytotoxicity assays produced different results depending on the washing time, washing solution, and 3D printed resin used (Figure 6). A three-way ANOVA revealed that cytotoxicity varied significantly with the washing time (F = 222.559, *p* < 0.001), washing solution (F = 76.683, *p* < 0.001), and material used (F = 249.099, *p* < 0.001). There were significant interactions between the material and washing solution (F = 69.965, *p* < 0.001); material and washing time (F = 9.104, *p* < 0.001); washing solution and washing time (F = 10.123, *p* < 0.001); and washing time, material, and washing solution (F = 13.240, *p* < 0.001). The cytotoxicity varied significantly with the washing time (Figure 6A). The cytotoxicity was highest in the control group (65.69 ± 22.55%), and there was a trend of a significant decrease as the washing time increased. The cytotoxicity was significantly lower for washing times of 15 min (9.52 ± 7.83%), 30 min (9.44 ± 9.66%), 60 min (5.86 ± 5.76%), and 90 min (5.04 ± 4.38%). The cytotoxicity was 24.08 ± 21.32% in the IPA washing group and 20.06 ± 25.42% in the TPM washing group (Figure 6B). The cytotoxicity was significantly higher in the ND group (26.71 ± 26.07%) than in the FL group (17.43 ± 19.64%) (Figure 6C).

Figure 7 shows the results of a one-way ANOVA used for identifying differences in cytotoxicity according to the washing time for each material and washing solution. In the ND-IPA group, the cytotoxicity was significantly higher for the washing times of the control (66.17 ± 11.67%) and 3 min groups (55.97 ± 12.27%), and significantly lower for the washing times of the 10 min (12.47 ± 4.11%), 60 min (8.84 ± 2.71%), and 90 min groups (9.35 ± 4.36%) (Figure 7A). In the ND-TPM group, the cytotoxicity was significantly higher for the washing time of the control group (96.58 ± 15.12%), and significantly lower for the washing times of the 15 min (10.02 ± 5.01%), 60 min (10.62 ± 7.28%), and 90 min groups (6.16 ± 3.77%) (Figure 7B). In the FL-IPA group, the cytotoxicity was significantly higher for the washing times of the control (50.10 ± 11.21%) and 3 min groups (41.61 ± 10.10%), and significantly lower for the washing times of the 30 min (5.57 ± 5.20%), 60 min (2.90 ± 3.28%), and 90 min groups (3.02 ± 2.93%) (Figure 7C). In the FL-TPM group, the cytotoxicity was significantly lower for all washing times except the control (49.93 ± 10.21%) and 3 min groups (32.59 ± 5.86%) (Figure 7D). We found that the cytotoxicity decreased with an increase in the washing time in all groups. All data for cytotoxicity after culturing for 24–72 h are listed in Table 4.

#### 3.1.3. CLSM Analysis

The morphology of cells attached to the specimen surfaces was observed using CLSM (Figure 8). Differences in the morphologies, sizes, and numbers of blue-stained multinucleate cells and green-stained cytoplasm and filopodia became evident with the increasing washing time. For the washing time used for the control group, there were adherent cells with very poor morphology in all groups, while the 3 min washing group showed small round cells with a few filopodia at the cell edge. In the 5 and 10 min washing groups, the cells had elongated or round shapes, and relatively few cells and nuclei remained. In the 15 min washing group, the cell cytoplasm mostly appeared elongated, with an increased number of cells. In the 30, 60, and 90 min washing groups, elongated fibroblasts were observed, as were large numbers of cells and cell-to-cell contacts. As the washing time increased, the stretched cytoplasm of the fibroblasts became conspicuous and the numbers of cells increased along with the contacts with neighboring cells.

### 3.2. Flexural Strength

A three-way ANOVA was performed to determine the effects of washing time, washing solution, and 3D printed resin material on flexural strength. The washing time (F = 2.109, *p* = 0.041), washing solution (F = 24.731, *p* < 0.001), and material used (F = 14.684, *p* < 0.001) were all found to significantly affect the flexural strength (Figure 9). The flexural strength was significantly higher in the CRT group (122.43 ± 16.05 MPa, mean ± standard deviation) than in the 60 min washing group (115.43 ± 9.75 MPa) (Figure 9A), significantly lower in the TPM washing group (115.78 ± 10.41 MPa) than in the IPA washing group (121.13 ± 12.91 MPa) (Figure 9B), and significantly lower in the ND group (120.48 ± 11.86 MPa) than in the FL group (116.43 ± 11.86 MPa) (Figure 9C). There was also a significant interaction effect between the washing solution and the material used (F = 7.520, *p* = 0.006). In addition, a one-way ANOVA was performed to verify the difference in flexural strength according to the washing times used for each 3D printed resin material and washing solution and revealed no significant difference in any of the groups.

### 3.3. Flexural Modulus

A three-way ANOVA was performed to determine the effects of the washing time, washing solution, and 3D printed resin material used on the flexural modulus. Washing time (F = 13.163, *p* < 0.001) and material (F = 257.151, *p* < 0.001) were found to significantly affect the flexural modulus, whereas the washing solution used (F = 0.100, *p* = 0.752) did not (Figure 10). There was a significant interaction between material and time (F = 4.753, *p* < 0.001) and between washing solution and time (F = 2.705, *p* = 0.09). The relationship between flexural modulus and washing time varied significantly between groups. The flexural modulus was significantly higher in the group without washing (1.30 ± 0.28 GPa) than in all the other washing groups except for the 3 min one, and it was significantly lower for the groups with washing times of 60 min (1.04 ± 0.25 GPa), 90 min (1.01 ± 0.21 GPa), and 15 min (1.09 ± 0.24 GPa) (Figure 10A). Overall, the flexural modulus tended to decrease as the washing time increased. The flexural modulus did not differ between the IPA washing group (1.12 ± 0.27 GPa) and the TPM washing group (1.13 ± 0.28 GPa) (Figure 10B), or between the ND (1.28 ± 0.24 GPa) and FL (0.98 ± 0.22 GPa) groups (Figure 10C). A one-way ANOVA performed to identify differences in flexural modulus according to the washing time for each 3D printed resin material and washing solution (Figure 11) revealed no significant difference in the ND group. In the FL-IPA group, the flexural modulus was highest in the control group (1.20 ± 0.21 GPa) and significantly lower for the washing times of the 5 min (0.95 ± 0.16 GPa), 60 min (0.79 ± 0.10 GPa), and 90 min groups (0.90 ± 0.13 GPa). In the FL-TPM group, the flexural modulus was significantly higher in the control group (1.33 ± 0.32 GPa), and did not differ between all of the other groups.

### 3.4. *SEM* Analysis

Four representative SEM images from the qualitative analysis (for washing times of the 5, 30, and 90 min groups and control group) are shown in Figure 12. The four groups of matte specimens demonstrated surface characteristics that varied with the material used. A low-magnification analysis at 40× magnification revealed that the surface of the ND resin produced by 3D printing using the DLP method showed a characteristic grid pattern, while the surface of the FL resin produced by 3D printing using the SLA method showed a relatively smooth surface. However, there were no differences according to the washing time used. The high-magnification analysis at 1000× magnification revealed differences in surface characteristics between groups, but no cracks or mechanical damage due to the washing process were observed.

## 4. Discussion

Crowns and dentures printed with dental 3D printed resin are in direct contact with soft and hard tissues in the oral cavity for a long time and must withstand large mastication forces. Accordingly, a high biocompatibility is needed and the mechanical properties need to be considered very carefully when selecting these materials. This study evaluated the effects of the use of various combinations of washing solutions, washing times, and materials. Our analyses of these three main parameters revealed partial differences according to the washing solutions and materials used, and observed that the washing times significantly affected the biological properties of the 3D printed samples. Therefore, the null hypothesis was partially accepted.

Figure 4A shows that cell viability was significantly affected by the washing time used. The polymerization of the methacrylate-based monomer of the resin composite results in the formation of a polymer with a highly cross-linked structure, but since the conversion of the monomer was not complete, significant numbers of unreacted double bonds remained [32]. The materials used in our study were methacrylate-based resins, including urethane dimethacrylate, propylidynetrimethyl trimethacrylate, and phosphine oxide. These components are known to exert cytotoxic and genotoxic effects when present as uncured monomers [33,34], and may lead to mutagenesis, DNA damage, or even cell death [34,35,36]. Previous studies have attempted to explain the effect of the cell contact of 3D printed resin in direct contact with soft tissues in the oral cavity [14,37,38]. Therefore, the direct approach of seeding the HGF on specimens was used. Kurzmann and colleagues applied an indirect approach, observing cytotoxicity by culturing cells in a normal well and placing a resin specimen on them, and found that the results were consistent with those of the direct approach of culturing cells directly on the specimen [38]. Additionally, as in the study of Kreß and colleagues, considering the long-term oral contact, cytotoxicity was observed every 24 h during a total culture period of 72 h, which contrasts with the standard exposure time of only 24 h (according to ISO 7405 [39]).

In our study, it is particularly interesting that the cell viability in all groups increased significantly as the washing time increased up to 90 min. This can be explained by an increase in the washing time increasing the removal of uncured monomers from the surface, leading to increased cell viability. Xu and colleagues reported that 5 min of ultrasonic washing with IPA increased cell viability, while increasing the washing time beyond 5 min did not further improve the cell viability [40]. Contrary to our study, it appears that the use of ultrasonic cleaners can rapidly increase cell viability. González and colleagues also reported that cell proliferation was more pronounced in a sample sonicated for 5 min than in a sample immersed in ethanol or acetone for 2 h, indicating that sonication was effective in reducing cytotoxicity [41]. Atay and colleagues classified a cell viability of greater than 90% as indicating that the substance is not cytotoxic, with values of 60–90% indicating mild cytotoxicity, 30–59% indicating moderate cytotoxicity, and less than 30% indicating severe cytotoxicity [42]. According to this classification, in the present study there were no significant differences between the washing solutions, while there were significant differences between the materials, but both materials could be classified as having moderate cytotoxicity (cell viability of 30–59%). It can also be suggested that more than 60 min of washing is required to reach a cell viability in excess of 60%. However, the results of Atay and colleagues were obtained in cytotoxicity evaluations of different CAD/CAM materials using extracts from culture media and not by direct contact [43]. Cytotoxicity decreased as the washing time increased in all of our groups, showing a similar trend to cell viability. In particular, there was a significant difference between the control group and the 90 min washing group, demonstrating that the washing process affects the biocompatibility of the material. A three-way ANOVA revealed that the cytotoxicity was significantly reduced after washing for more than 15 min. In particular, there was little difference between washing times of 60 and 90 min in all groups, for which a large decrease in cytotoxicity was observed. In particular, the cytotoxicity was very high in the control and 3 min washing groups, and so washing for at least 5 min seems to be necessary. Although there were differences between the washing solutions and materials used, all groups showed less than 30% cytotoxicity, and so the washing process after manufacturing seems to be effective in reducing the number of unreacted monomers to below the cell tolerance threshold.

Fibroblasts are the most abundant cells present in connective tissue and directly guide and regulate cellular functions via receptor–ligand interactions [44]. They also communicate with other cells by secreting or releasing growth factors and cytokines from the extracellular matrix and develop cell-to-cell contacts with each other as well as other types of cells to create complex cellular communication networks [44]. Fibroblasts also regulate tissue development, organogenesis, homeostasis, and maintenance, and play key roles in various physiological and pathological situations, including wound healing, inflammation, and cancer [45]. Therefore, it is important to verify the fibroblast cytotoxicity of 3D printed resins. Our CLSM images revealed clear differences and trends in the size, morphology, and number of cells with washing time, similar to the results obtained in the cell viability and cytotoxicity tests. As the washing time increased, the number of normal cells increased, and elongated fibroblasts and cell-to-cell contact were observed. Prominently, large numbers of normal cells were found on the surface of the specimens in the 30, 60, and 90 min washing groups compared with the other groups. Moreover, there were almost no normal adherent cells in the control and 3 min washing groups, and only a few very poorly formed filopodia were observed. Unwashed monomers remaining on the surface can result in irreversible disturbances of basic cellular functions such as cell proliferation, enzymatic activity, cell morphology, membrane integrity, cell metabolism, and survival [34]. Therefore, it is important to determine the optimal washing time for meeting the biological requirements of 3D printed resins. The results of this study suggest that the biocompatibility of products constructed from 3D printed resins improves as the washing time increases up to 90 min.

Three-way ANOVAs of all the experimental groups included in this study revealed that the flexural strength was significantly higher in the control group, but that there was no significant tendency for the flexural strength to decrease as the washing time increased. Xu and colleagues similarly found no difference in flexural strength for washing times of 5 to 60 min [40]. However, those authors found that the flexural strength decreased after washing for more than 12 h, suggesting that prolonged washing affects the flexural strength due to the solvent molecules penetrating from the surface into the resin matrix, causing the polymer network to relax [40]. Our analysis of the flexural strength according to the use of different washing solutions revealed that the flexural strength was significantly higher in the IPA washing group than in the TPM washing group. However, it is difficult to say that the difference was clinically significant, since it was less than 10 MPa. Mayer and colleagues recorded the highest degree of conversion after washing temporary 3D printed fixed dental prosthesis materials with butyl glycol and IPA, and that the flexural strength was lowest after washing with acetone and butyl glycol [46]. These observations suggest that flexural strength is affected by differences in washing solutions.

Unlike flexural strength, flexural modulus showed a tendency to decrease with increasing washing time (Figure 9). A one-way ANOVA used to evaluate washing time within groups revealed that as the washing time increased in the FL group, the flexural modulus tended to decrease. Awada and colleagues reported that a polymer-based material had a relatively low flexural modulus compared with its high flexural strength. This can be explained by the increase in the ability of elastic materials to withstand loads as they undergo elastic deformation before they break [47]. After being stored in ethanol, the Wallace hardness increases significantly compared with that after storage in air, which is due to the resin-softening effect of ethanol [48]. Therefore, in this study, as the washing time increased, the molecular weight of the resin decreased and the flexural modulus decreased, whereas it is thought that the ability of the flexible material to bear loads increased as it undergoes elastic deformation. As a result, it appears that the flexural modulus decreased with the increasing washing time, but that the flexural strength might not have. As flexural strength increases, the modulus of resilience increases, and as flexural modulus increases the modulus of resilience decreases. Therefore, the modulus of resilience directly depends on the interaction between flexural strength and modulus [47]. As shown in our experimental results, a decrease in flexural modulus without a change in flexural strength means an increase in the modulus of resilience. The modulus of resilience is the amount of strain energy per unit volume that a material can absorb without permanent deformation. An increase in this value means an increase in the elasticity of the material.

The flexural modulus did not differ between the washing solutions, and FL showed a low flexural modulus, which may be due to urethane dimethacrylate being one of its components. Urethane dimethacrylate is a monomer that reportedly exhibits lower viscosity and greater flexibility due to the presence of urethane bonds, and it is known that resin composites based on this monomer exhibit greater toughness [49]. However, limitations associated with patents mean that manufacturers do not disclose the detailed monomer and polymer components of their products, which makes it difficult to analyze differences in results based on specific compounds. The ISO 4049 standard for resin composites specifies only the requirement of the minimum flexural strength value for the mechanical properties. However, the elastic modulus of resin composites can have more clinical relevance, since it provides the resistance of material to deformation, which is correlated with the clinical performance [50]. The flexural modulus is a useful measure for determining whether a material will break or rupture due to applied stress. A lower flexural modulus lowers the bending resistance and can cause problems such as the repeated elastic deformation of a resilient restoration’s margin, which might lead to microleakage [47]. Furthermore, a low flexural modulus could lead to restorative deformation under load, resulting in accelerated wear and hoop stress that may cause crown debonding [51,52]; therefore, it is considered that washing for a long time should be avoided. However, a material with a high strength and low flexural modulus is suitable for dental trauma management because a flexible splint assists in healing [53]. Therefore, it seems that the washing time should be considered according to the purpose of use of the dental prosthetics.

The SEM images in Figure 12 show that washing with IPA for 90 min did not allow sufficient solvent to penetrate so as to cause irreversible deformation. In particular, TPM has a lower immersion sensitivity to cured resin than IPA does, and so washing for up to 90 min using both washing solutions did not seem to affect the surface properties. Xu and colleagues also reported that washing with IPA for up to 60 min did not cause any surface changes [40], which is consistent with the results of our study. However, cracks were found on the surface after washing for more than 12 h, indicating that prolonged immersion could result in irreversible deformation. This has been attributed to surface expansion, with solvent molecules being absorbed into the polymer network on the outermost surface to increase the volume, inducing various stresses and osmotic pressures [40]. According to the above experimental results, washing for more than 90 min can reduce the flexural modulus, and so this washing time should be considered the upper limit. In addition, the flexural strength and surface defects of 3D printed resin seem to be more affected by aging and the printing direction [54,55,56], surface treatment [57,58], printing method [59], and post-curing temperature and time [14,60] than by any changes to the washing time within the upper limit of 90 min. Additionally, in order to more clearly confirm the effect of washing time on mechanical properties, it is considered that a follow-up study with a control group of other materials or conditions is needed.

IPA is the most common solvent used to wash unpolymerized resin from the surface of 3D printed restorations. IPA solvent evaporates due to its high vapor pressure and strong volatility. IPA is also flammable and is classified as a toxic substance that can also have negative effects on users; hence, it is a liquid that requires certain safety precautions to be taken [16,61]. In contrast, TPM is a nonflammable, nontoxic liquid with a low vapor pressure, and does not require the safety standards associated with IPA, but it does require additional washing with a brush or detergent and is more expensive than IPA [16]. According to the results obtained in the present study, both of these washing solutions can improve biocompatibility without significantly impairing the mechanical properties of the material. Mostafavi and colleagues investigated the effects of washing with TPM and IPA on the manufacturing accuracy of vat-polymerized dental materials and found that both the trueness and precision were higher for TPM than for IPA [16]. Therefore, our experimental results can be used as a basis for selecting a washing solution that suits specific situations and requirements. In addition, manufacturers suggest washing 3D printed objects for longer than the recommended time if they are large or deep and narrow, but there are no exact guidelines for this. According to the results of the present study, washing for up to 90 min does not cause critical defects in mechanical properties, which can help when determining the optimal washing time according to the shape and size of specific objects. Furthermore, as a result of the biocompatibility test carried out in this study, cytotoxicity was greatly reduced when washing for more than 15 min, and most of the cell morphology was maintained normally. Additionally, the modulus decreased as the washing time increased, but in most groups, 15 min of washing did not cause any critical defects. Therefore, it is possible to reduce unnecessary washing time beyond that by recommending a minimum washing time of 15 min in object processing.

One limitation of this study was the use of a single type of washer and washing method. The use of different types of ultrasonic equipment or solutions that enable the temperature control of the solvent may produce different results. Another limitation was that the biocompatibility and mechanical testing did not reproduce the oral environment, including the effect of acid and base foods, saliva, etc. Moreover, the accuracy of a 3D printed part finally produced by focusing only on mechanical and biological properties may vary depending on various parameters.

## 5. Conclusions

This study investigated the effects of washing time and washing solution on the mechanical properties, biocompatibility, and surface characteristics of 3D printed resins. Within the limitations of our study, the following conclusions can be drawn:(1)As the washing time increased, the cell viability increased and the cytotoxicity decreased, indicating an improved biocompatibility;(2)As the washing time increased, there were no reductions in the flexural strength or changes in surface defects, while the flexural modulus decreased;(3)Neither washing solution (IPA or TPM) caused significant defects in mechanical properties, and the biocompatibility increased with the washing time for both solutions.

The results reported here will be helpful for determining an efficient workflow that can be used in clinical practice by providing a guide to appropriate washing solutions and washing times to increase biocompatibility without affecting the mechanical strength of 3D printed dental resin materials.

## Figures and Tables

**Figure 1 polymers-13-04410-f001:**
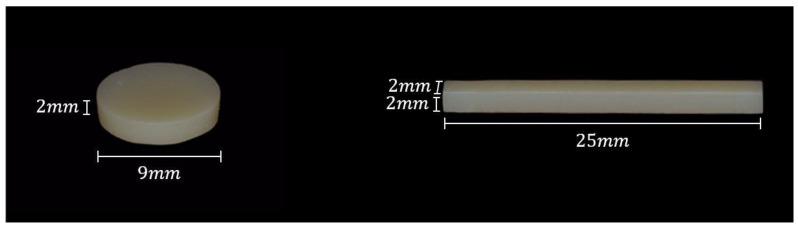
Additive Manufacturing 3D Printed Specimens.

**Figure 2 polymers-13-04410-f002:**
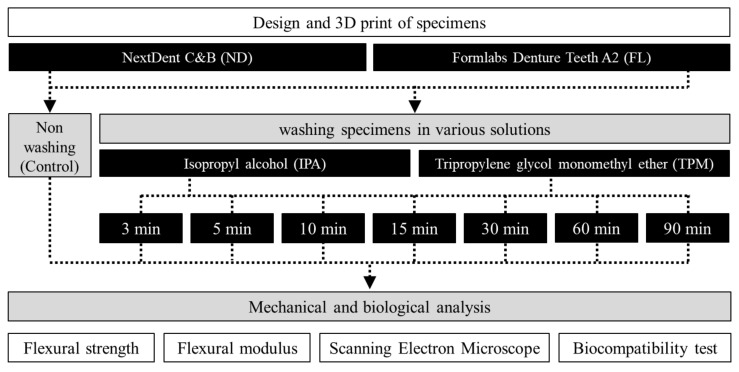
Workflow of the overall experimental process, showing the washing conditions and types of experiment carried out.

**Figure 3 polymers-13-04410-f003:**
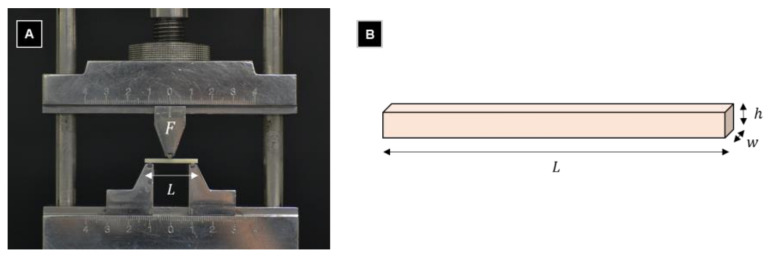
Image (**A**) and schematic (**B**) of the test setup for SO 10447.

**Figure 4 polymers-13-04410-f004:**
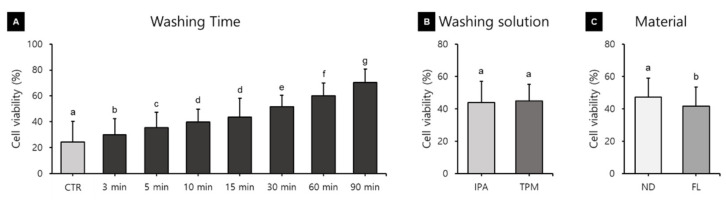
Results from a three-way ANOVA regarding cell viability according to (**A**) the washing time, (**B**) the washing solution, and (**C**) the 3D printed resin material (24–72 h). Data are the mean and standard deviation values of 5 3D printed specimens. Different lower-case letters indicate significant differences.

**Figure 5 polymers-13-04410-f005:**
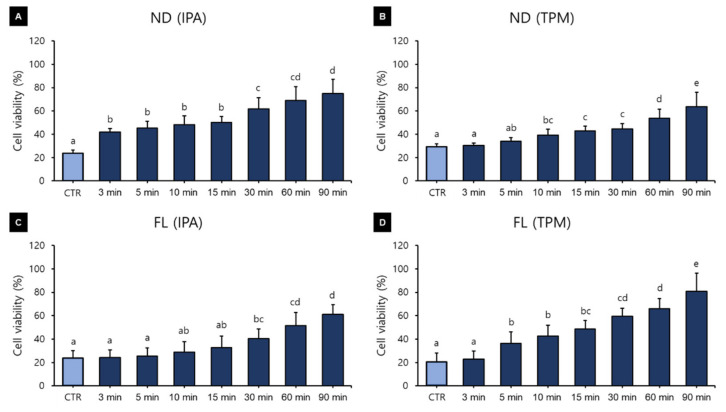
Results from a one-way ANOVA regarding cell viability according to the washing time for each 3D printed resin and washing solution (24–72 h): (**A**) ND-IPA, (**B**) ND-TPM, (**C**) FL-IPA, and (**D**) FL-TPM groups. Data are the mean and standard deviation values of 5 3D printed specimens. Different lower-case letters indicate significant differences.

**Figure 6 polymers-13-04410-f006:**
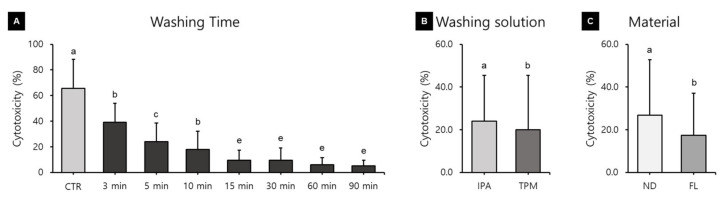
Results from a three-way ANOVA regarding cytotoxicity according to (**A**) the washing time, (**B**) the washing solution, and (**C**) the 3D printed resin material used (24–72 h). Data are the mean and standard deviation values of 5 3D printed specimens. Different lower-case letters indicate significant differences.

**Figure 7 polymers-13-04410-f007:**
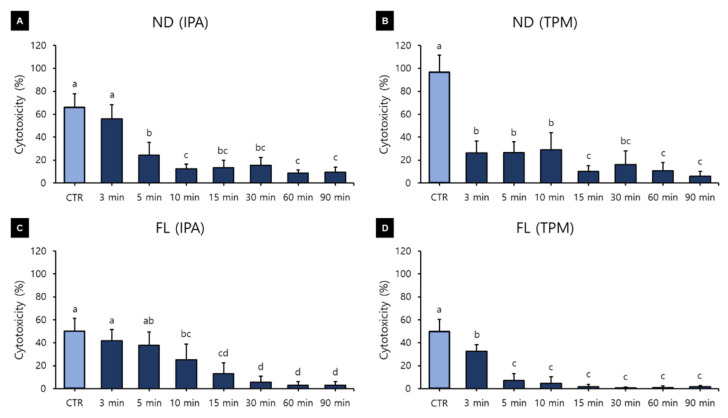
Results from a one-way ANOVA regarding cytotoxicity according to the washing time used for each 3D printed resin and washing solution (24–72 h): (**A**) ND-IPA, (**B**) ND-TPM, (**C**) FL-IPA, and (**D**) FL-TPM groups. Data are the mean and standard deviation values of 5 3D printed specimens. Different lower-case letters indicate significant differences.

**Figure 8 polymers-13-04410-f008:**
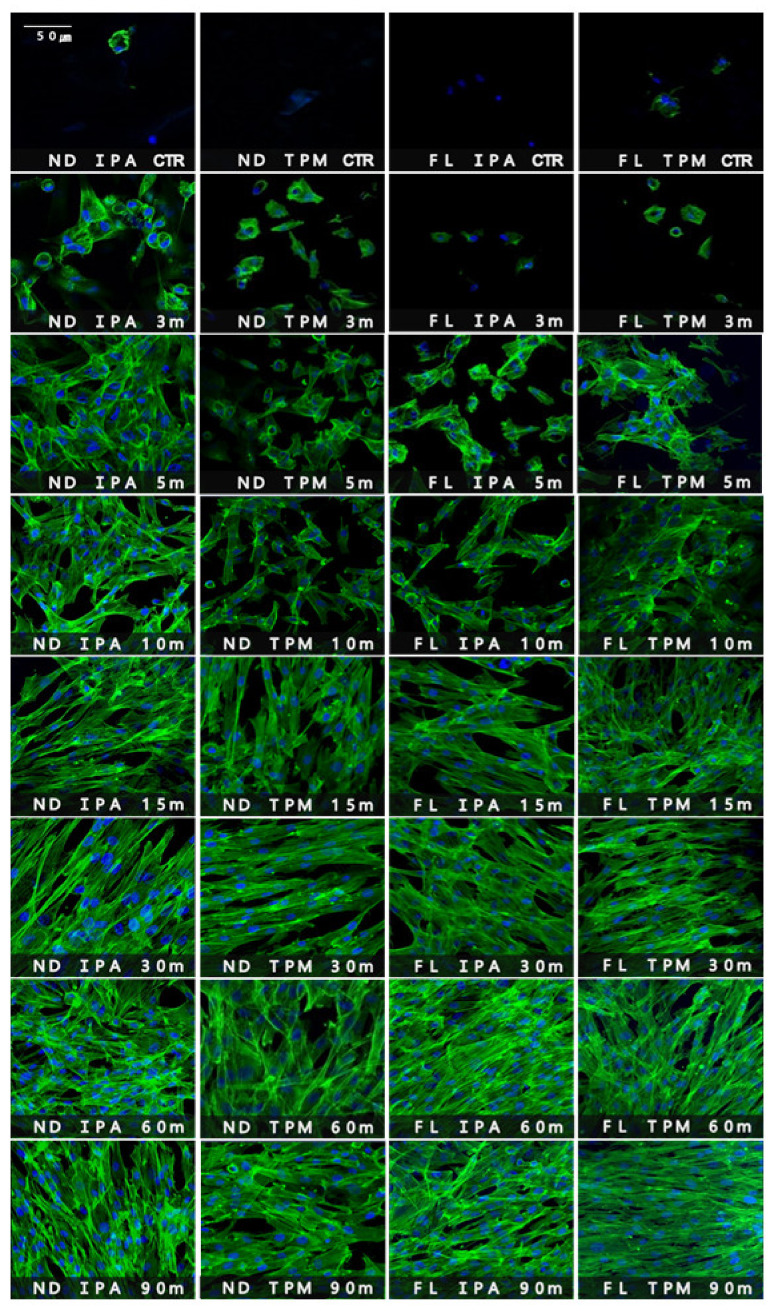
Confocal laser scanning microscopy images of human gingival fibroblasts cultures (5 × 10^4^ cells/mL) on 3D printed specimens washed with various solutions and for various times after 24 h of incubation.

**Figure 9 polymers-13-04410-f009:**
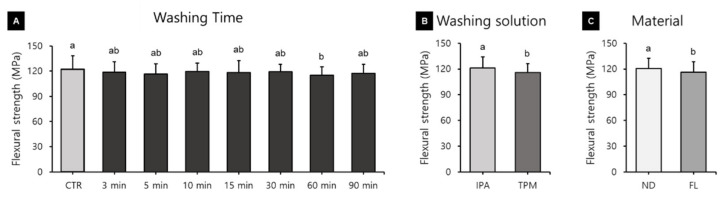
Results from a three-way ANOVA carried out to evaluate flexural strength according to (**A**) the washing time, (**B**) the washing solution, and (**C**) the 3D printed resin material used. Data are the mean and standard deviation values for 15 3D printed specimens. Different lower-case letters indicate significant differences.

**Figure 10 polymers-13-04410-f010:**
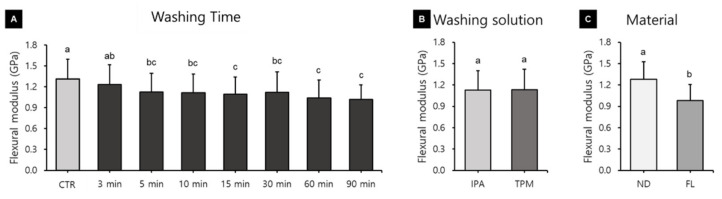
Results from a three-way ANOVA used to evaluate flexural modulus according to (**A**) the washing time, (**B**) the washing solution, and (**C**) the 3D printed resin material used. Data are the mean and standard deviation values of 5 3D printed specimens. Different lower-case letters indicate significant differences.

**Figure 11 polymers-13-04410-f011:**
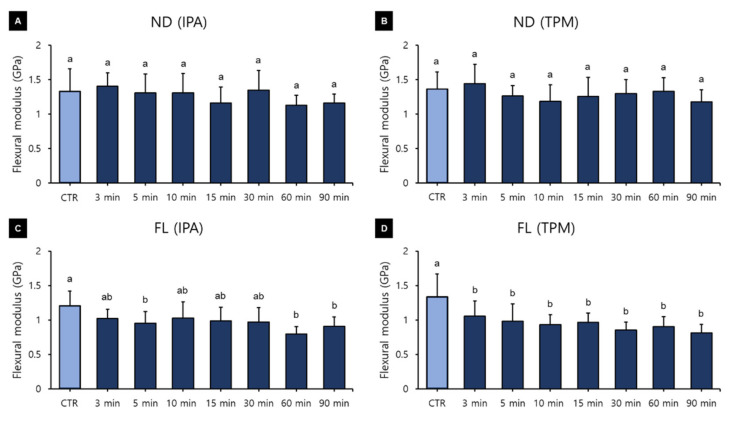
Results from a one-way ANOVA used to evaluate flexural modulus according to the washing time used for each 3D printed resin and the washing solution: (**A**) ND-IPA, (**B**) ND-TPM, (**C**) FL-IPA, and (**D**) FL-TPM groups. Data are the mean and standard deviation values of 5 3D printed specimens. Different lower-case letters indicate a significant difference.

**Figure 12 polymers-13-04410-f012:**
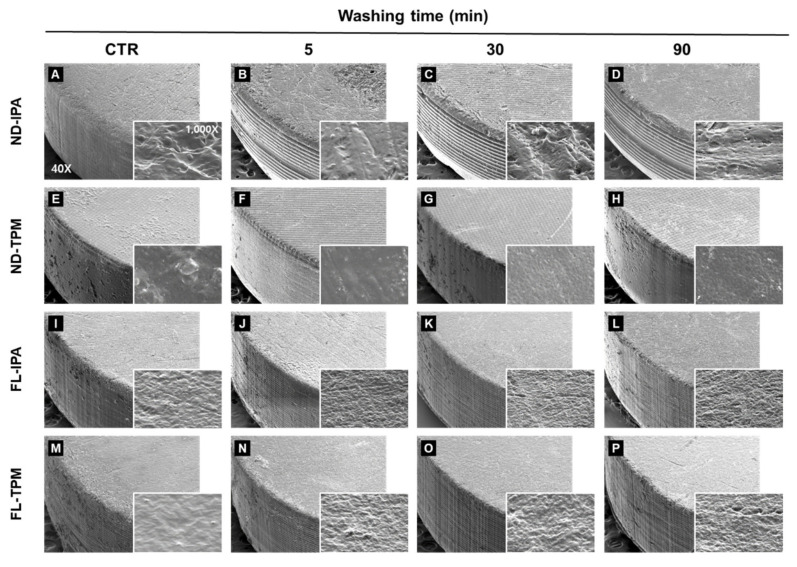
Qualitative scanning electron microscopy images of the surface topography at 40× and 1000× in four representative groups for washing times of 5, 30, and 90 min and the control group: ND-IPA (**A**–**D**), ND-TPM (**E**–**H**), FL-IPA (**I**–**L**), and FL-TPM (**M**–**P**) groups.

**Table 1 polymers-13-04410-t001:** The materials used for 3D printing in this study.

Product	Code	Composition	Manufacturer
Formlabs Denture Teeth A2	FLs	Methacrylate monomer, urethane dimethacrylate, propylidynetrimethyl trimethacrylate, diphenyl(2,4,6-trimethylbenzoyl) phosphine oxide	Formlabs, Somerville, MA, USA
NextDent C&B	ND	>90% methacrylic oligomers, methacrylate monomer, <3% phosphine oxides, pigment	NextDent, Soesterberg, the Netherlands

**Table 2 polymers-13-04410-t002:** Characteristics of the two washing solutions.

Product	Code	Purity	Evaporation Rate (n-Butyl Acetate = 1)	Flash Point (°C)	Vapor Pressure (mmHg)
Isopropyl alcohol	IPA	90%	0.0026	12 °C	45.4
Tripropylene glycol monomethyl ether	TPM	<95%	2.3	111 °C	> 1

**Table 3 polymers-13-04410-t003:** Cell viability after 24, 48, and 72 h in each group according to the washing time used for each 3D printed resin and the washing solution.

Group	Cultivation Time (h)	Washing Time (min)							
		CTR	3	5	10	15	30	60	90
ND- IPA	24	24.06 ± 3.48	42.82 ± 3.48	42.64 ± 3.36	43.37 ± 5.78	49.20 ± 5.62	57.23 ± 5.58	66.92 ± 10.98	77.23 ± 16.40
	48	23.94 ± 1.01	43.62 ± 1.28	46.47 ± 4.84	53.00 ± 6.97	52.37 ± 5.00	66.61 ± 13.69	75.21 ± 13.34	81.00 ± 5.40
	72	23.08 ± 3.63	39.53 ± 1.73	46.39 ± 9.02	48.75 ± 7.22	49.47 ± 4.29	61.77 ± 5.78	65.31 ± 10.85	66.04 ± 9.02
ND- TPM	24	28.05 ± 1.68	28.59 ± 1.45	32.21 ± 2.56	35.96 ± 0.67	39.51 ± 0.41	43.02 ± 4.48	54.25 ± 6.87	64.14 ± 13.35
	48	31.85 ± 1.19	32.84 ± 1.20	35.82 ± 2.34	42.14 ± 6.84	47.19 ± 2.76	47.50 ± 5.31	57.24 ± 10.79	68.16 ± 15.43
	72	28.52 ± 1.22	29.71 ± 0.88	34.25 ± 2.85	39.89 ± 3.35	42.20 ± 2.15	43.26 ± 3.24	49.85 ± 1.20	59.15 ± 6.13
FL- IPA	24	24.60 ± 0.50	25.52 ± 1.75	28.28 ± 2.11	32.84 ± 4.25	38.39 ± 4.77	43.16 ± 5.05	57.82 ± 7.35	59.53 ± 4.40
	48	30.91 ± 0.52	31.07 ± 0.27	31.96 ± 0.81	36.20 ± 3.32	40.11 ± 2.49	45.69 ± 3.74	54.78 ± 7.92	57.43 ± 5.39
	72	15.70 ± 0.37	15.78 ± 0.29	16.09 ± 0.20	17.78 ± 1.36	19.73 ± 1.17	32.23 ± 7.82	41.83 ± 12.54	66.49 ± 11.49
FL- TPM	24	28.04 ± 1.01	29.37 ± 2.35	38.27 ± 7.48	44.00 ± 13.40	47.00 ± 11.80	58.78 ± 7.90	66.32 ± 9.64	71.98 ± 13.73
	48	22.94 ± 0.31	24.96 ± 0.89	39.44 ± 10.46	45.57 ± 5.34	50.13 ± 4.26	58.77 ± 8.08	63.68 ± 10.94	94.64 ± 16.43
	72	10.96 ± 0.16	13.99 ± 3.61	31.80 ± 11.28	38.75 ± 6.60	48.67 ± 5.02	60.59 ± 6.44	68.31 ± 4.35	75.78 ± 2.65

Data are the mean and standard deviation values of the percentage of cell viability (24~72 h).

**Table 4 polymers-13-04410-t004:** Cytotoxicity after 24, 48, and 72 h in each group according to the washing time used for each 3D printed resin and washing solution.

Group	Cultivation Time (h)	Washing Time (min)							
		CTR	3	5	10	15	30	60	90
ND- IPA	24	57.38 ± 1.61	43.59 ± 4.07	16.79 ± 2.81	10.40 ± 1.18	10.20 ± 1.07	11.94 ± 1.86	10.44 ± 1.11	9.77 ± 1.49
	48	62.97 ± 12.83	54.01 ± 6.29	18.48 ± 6.59	9.59 ± 1.34	9.69 ± 2.56	11.04 ± 3.48	5.97 ± 2.32	6.42 ± 3.38
	72	78.17 ± 4.53	70.33 ± 4.09	37.34 ± 8.32	17.42 ± 3.11	20.18 ± 7.18	23.21 ± 5.31	10.14 ± 1.87	11.85 ± 5.86
ND- TPM	24	95.35 ± 18.17	14.52 ± 6.19	14.75 ± 2.63	14.29 ± 5.22	5.56 ± 2.44	6.05 ± 5.21	3.57 ± 3.08	2.23 ± 1.39
	48	96.22 ± 15.73	30.93 ± 8.03	32.03 ± 2.63	33.11 ± 14.06	10.41 ± 4.74	17.06 ± 12.72	11.57 ± 3.81	5.88 ± 1.71
	72	98.18 ± 14.77	33.18 ± 5.57	33.49 ± 3.44	39.91 ± 9.66	14.11 ± 3.69	25.27 ± 7.46	16.74 ± 7.18	10.37 ± 1.86
FL- IPA	24	40.70 ± 5.46	32.54 ± 1.49	27.03 ± 3.79	11.42 ± 1.51	4.76 ± 0.91	1.57 ± 0.74	0.06 ± 0.45	0.68 ± 0.39
	48	45.73 ± 5.49	37.26 ± 1.30	33.27 ± 1.00	22.01 ± 1.63	9.28 ± 2.48	2.90 ± 1.35	1.93 ± 0.90	1.97 ± 0.97
	72	63.89 ± 2.86	55.04 ± 1.15	53.28 ± 2.01	42.48 ± 2.38	25.30 ± 4.19	12.26 ± 2.78	6.73 ± 2.70	6.42 ± 2.53
FL- TPM	24	40.18 ± 2.21	27.34 ± 1.90	3.52 ± 3.43	1.62 ± 1.38	0.17 ± 1.94	0.49 ± 0.84	0.59 ± 1.03	1.09 ± 0.62
	48	46.63 ± 1.83	30.46 ± 0.62	4.95 ± 2.69	3.14 ± 2.56	1.69 ± 1.52	1.40 ± 0.22	2.06 ± 1.04	2.41 ± 0.69
	72	62.98 ± 3.45	39.98 ± 2.79	13.06 ± 5.75	9.34 ± 7.65	2.92 ± 2.12	0.18 ± 0.50	0.62 ± 0.71	1.38 ± 1.12

Data are the mean and standard deviation values of the percentage cytotoxicity (24~72 h).

## Data Availability

The data presented in this study are available upon request from the corresponding author.

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
