# Peer review of "Effects of the Washing Time and Washing Solution on the Biocompatibility and Mechanical Properties of 3D Printed Dental Resin Materials"

_polymers, 2021, doi:10.3390/polym13244410_

Round 1

Reviewer 1 Report

The authors again chose not to answer the comments appropriately. I suggest rejecting the work.

Author Response

보고서 1

저자들은 몇 가지 3D 인쇄 치과용 수지 재료에 대한 훌륭한 연구를 수행했습니다. 주제가 흥미롭고 좋은 실험이 선택됩니다. 그러나 출판을 제안하기 전에 답변해야 할 몇 가지 우려 사항이 있습니다.

의견: 영어 및 스타일의 광범위한 편집이 필요합니다.

답변: https://www.mdpi.com/authors/english에 나열된 편집 서비스 중 하나를 사용하여 편집했습니다.

< 이전 개정 주석 >

Comment: The main problem I see rises from the fact that, except for the biocompatibility results, almost all tests show similar results for various conditions. It means various situations almost did not affect the 3D-printed materials. I suggest adding one or two other materials/conditions as control(s). Which materials are affected? Which conditions affect the printed materials? Please add them to the figures. Flat curves in most of the plots do not provide sufficient data for the reader(s).

Answer: Thank you for your good suggestion. As you suggested, we checked various factors in order to confirm the change of biocompatibility and mechanical properties according to the washing time and two type washing solution. Two materials were tested, the first material being Formlabs Denture Teeth and the second material being NextDent Crown & Bridge. As a result, it was confirmed that as the washing time increased, both of the tested materials were significantly affected by biocompatibility and increased. This suggests that biocompatibility, which is a very important factor in materials that are mounted directly into the patient's mouth, was successfully increased with increasing washing time. In addition, three tests were conducted to confirm whether the mechanical properties were affected by the washing time and solution. As a result, it was confirmed that the flexural modulus was significantly affected by the washing time. This proves that excessive washing time should be avoided as it is an important factor in determining micro-leakage and crown detachment due to marginal deformation in the final prosthesis[1, 2].

As your comment, in this study, the effect of washing solution and washing time on materials widely used in clinical practice was confirmed, and as a result, the optimal time and solution were confirmed. Through this, we provide guidelines to configure an efficient workflow for clinicians and dental technicians.

Comment: The authors have not answered the comments appropriately. As I mentioned in the last review report, as the MAIN problem with the work, they need to include some test results to make the charts valuable. Now, the manuscript is full of flat curves, from which not much data can be gained. The authors have actually skipped answering this important problem.

Answer: As you mentioned, we tried to help readers understand by adding an explanation of the biocompatibility results that showed a significant difference. In addition, the reason for the significant decrease in the flexural modulus was additionally explained, suggesting that the clinical importance of the flexural modulus and excessive washing time should be avoided.

Comment: Please provide figures of the printed parts. Authors need to add the real figures of the printed parts. Just providing a schematic does not help.

Answer: Thank you for your good comment. We have added the figures of the printed parts on Figure 1. Following your comment.

Figure 1. Additive manufacturing of the samples. (A) and (C) The size of the printed parts schematic.; (B) and (D) The 3D printed specimens (Red scale: 10 mm).

Comment: Can authors also include the effect of some (acid and base) foods, saliva, etc. on the printed parts? Authors also have skipped the suggestion of including the effect of food, which is super important for this study.

Answer: Thank you for your kind review of the content. As you mentioned, we have added the limitation of this study, that this in vitro study does not include the effect of some (acid and base) foods, saliva, etc. In discussion on line 698-700 and highlighted in red.

4. Discussion

One limitation of this study was the use of a single type of washer and washing method. Different types of ultrasonic equipment or solutions with temperature control of the solvent may produce different results. Another limitation was that the biocompatibility and mechanical testing did not reproduce the oral environment include the effect of acid and base foods, saliva, etc. Moreover, the accuracy of a 3D-printed part finally produced by focusing only on mechanical and biological properties may vary depending on various parameters.

Reference

  1. Awada, A. and D. Nathanson, Mechanical properties of resin-ceramic CAD/CAM restorative materials. The Journal of prosthetic dentistry, 2015. 114(4): p. 587-593.
  2. Jun, S.-K., et al., 수지복합체의 서로 다른 기계적 성질 사이의 상관관계 조사. 치과재료학회지, 2013. 32 (1): p. 48-57.

Reviewer 2 Report

The present manuscript is well organized and developed. I consider that it can be published in the current version.

Author Response

Thank you for your kind review of the content. 

Reviewer 3 Report

Why was three-way ANOVA used? It seems that the authors were trying to test correlations between some variables, and for that they used three-way ANOVA?

Moreover, it is not clear how many samples were analyzed in each group, for each time point. Please provide the number of samples for each comparison being made.

They mentioned that they checked normality with the Kolmogorov-Smirnov test, but regardless of that they opted for parametric tests from the start.

Author Response

Report 3

Comment: Why was three-way ANOVA used? It seems that the authors were trying to test correlations between some variables, and for that they used three-way ANOVA?

Answer: Thank you for your kind review of the content. In this study, 8 washing times including two materials, two washing solutions and controls were selected as factors. Samples were prepared by washing 2 types of resins in 2 washing solutions for 8 types different time points. Therefore, it was decided that these three variables cannot be viewed separately. In this study, 3-way was executed to identified the interaction between three variables and to check the effect of washing time, washing solution and materials on biocompatibility and mechanical properties. We have added explanations to help readers understand on the line 257-268 of statistical analysis and highlighted in red.

Comment: Moreover, it is not clear how many samples were analyzed in each group, for each time point. Please provide the number of samples for each comparison being made.

Answer: Thank you for your good comment. For all experiments (Flexural strength, Flexural modulus, Cell viability assay and Cytotoxicity assay), 15 samples per group of 4 experimental group (ND-IPA, ND-TPM, FL-IPA, FL-TPM) * for 8 types different time points were analyzed, and a total of 480 samples were analyzed. The number of samples is indicated on the line 175-176, 257-268 of Materials and Methods and statistical analysis and highlighted in red.

2.3. Biocompatibility test

2.3.1. In vitro cell culture and cell line

The plates were incubated for 24, 48 and 72 hours at 37°C and 5% CO2 before analyzing cell viability (n=15 per group), and cytotoxicity (n=15 per group).

Comment: They mentioned that they checked normality with the Kolmogorov-Smirnov test, but regardless of that they opted for parametric tests from the start.

Answer: Thank you for your good comment. As your comment, we thought that Shapiro-Wilk was suitable for the number of samples in this study, so we performed both normality tests with Kolmogorov-Smirnov test and Shapiro-Wilk test, compared and confirmed, and selected Shapiro-Wilk's result. However, in the raw data, variable transformation was performed by confirming that some variables do not follow the normal distribution, and normality was confirmed through the results of the normality test of the above data (p > .05). In addition, since the extracted sample data was judged to be sufficiently representative of the population with the statistical basic assumption of normality after variable transformation, a 3-way ANOVA test, a parametric analysis method based on the parametric assumption, was used. Al so, since the Kruskal-wallis test, a non-parametric analysis method, is limited in identifying the characteristics of the three interactions of material, solution, and time, 3-way ANOVA test and post-test were used. Finally, after transforming the variable, the Shapiro-Wilk test was performed and ANOVA was performed again with the data to confirm normality, and the F value and p value of the result (278-283, 361-365, 441-449, 464-467) were supplemented and highlighted in red.

2.6. Statistical analysis

The normality of all the data was confirmed by performing a Shapiro–Wilk test. Data from the cell viability (n=480) and cytotoxicity (n=480) assays were analyzed by three-way ANOVA using standard statistical software (version 25.0, SPSS Statistics, IBM, Armonk, NY, USA), followed by the Bonferroni and Scheffe tests. Through this, the interaction between washing time, washing solution, and materials and the effect of variables on biocompatibility were confirmed. A one-way ANOVA was also used to confirm differences in the experimental results according to the washing time within each group (α < 0.05). In order to confirm the effect of washing time, washing solution and material on mechanical properties, data on flexural strength(n=480) and flexural modulus(n=480) were analyzed using three-way ANOVA, followed by Bonferroni test and Tukey multiple-comparison test. In addition, a one-way ANOVA was used to confirm the differences according to the washing time within various groups.

Round 2

Reviewer 1 Report

The explanations provided convinced me. I suggest publishing this manuscript in this journal. The only two suggestions that I have are:

  • Please provide a better figure 1. Remove panels A and C, and use better real pictures. The ones that are currently used can be replaced with better pictures which show better the printed parts.
  • Replace Figure 3 with a real image.

Reviewer 3 Report

The manuscript now seems to be suitable for publication.

This manuscript is a resubmission of an earlier submission. The following is a list of the peer review reports and author responses from that submission.

Round 1

Reviewer 1 Report

There is lack of information regarding the effect of different washing solutions and washing times on the biocompatibility of the resulting resin restorations. Therefore the present study is original and relevant.

Abstract:

The acronyms meaning should be present.

Introduction:

Well written and contemporary.

Methods:

Did the disk-shaped samples followed any ISO parameter?

Why 50 mm printing layer thickness has been chosen?

Insert he reference that have determined the parameters applied in the topic 2.4

Discussion:

What is the interaction between flexural strength and flexural modulus? And what means increase the values of one of these properties while the other is still the same? Discuss it and how it can affect your materials behavior in the clinical applicability.

Improve your discussion with regard to the saving time during the printed object processing.

Author Response

Report 1

There is lack of information regarding the effect of different washing solutions and washing times on the biocompatibility of the resulting resin restorations. Therefore the present study is original and relevant.

Comment: The acronyms meaning should be present.

Answer: Thank you for your kind review of the content. I wrote about 3D acronyms in line 12 of abstract and edited the sentence to be concise. Also, in lines 15-18 of the abstract, washing solution acronyms are indicated and unnecessary acronyms have been removed for better understanding. The corrected parts was emphasized with red.

Abstract: Three-dimensional (3D) printing technology is highly recognized in the field of dentistry. 3D printed resin restorations must undergo a washing process to remove residual resin on the surface after manufacturing. However, the effect of different washing solutions and washing times on the biocompatibility of the resulting resin restorations is unclear. Therefore, we prepared 3D printed denture teeth and crown & bridge resin, then washed with 2 washing solutions (Isopropyl alcohol and Tripropylene glycol monomethyl ether) using different time points (3, 5, 10, 15, 30, 60, 90 minutes), then flexural strength, flexural modulus, and surface SEM imaging were analyzed. We also evaluated cell viability, cytotoxicity, and the status of human gingival fibroblasts using confocal laser scanning. Increasing the washing time did not significantly affect the flexural strength and surface, but the flexural modulus of the 90-minute washing group was 1.01 ± 0.21GPa (mean ± standard deviation), which was lower than that of all other groups, and decreased as the washing time increased. Increasing the washing time increased cell viability and decreased cytotoxicity (p < 0.001). Confocal laser scanning showed distinct differences in the morphology and number of fibroblasts. This study has confirmed that the washing time affected the biocompatibility and mechanical properties of dental 3D printed resins.

Comment: Did the disk-shaped samples followed any ISO parameter?

Answer: Thank you for your kind review.he ISO for the parameters of 3D printing resin samples has not yet been defined. Specimens were prepared by referring to ISO 7405:2009/(R)2015, Dentistry — Polymer-based restorative materials. According to 6.1.2 of ISO 7405 [1], the product was manufactured according to the manufacturer's protocol, and it was printed so that there was no deviation between the specimens.

6.1.2 General recommendations for sample preparation

For the preparation of test samples, consult the respective product standards and/or the manufacturer’s instructions, and follow those descriptions as closely as possible. Justify any deviation from the manufacturer's instructions. A detailed description of the sample preparation shall be included in the test report.

Comment: Why 500㎛ printing layer thickness has been chosen?

Answer: Thank you for your kind review of the content. According to Ian Gibson, I.G. Additive manufacturing technologies 3D printing, rapid prototyping, and direct digital manufacturing [2], The nominal layer thickness for most machines is around 0.1 mm. However, it should be noted that this is just a rule of thumb. For example, the layer thickness for some material extrusion machines is 0.254 mm, whereas layer thicknesses between 0.05 and 0.1mm are commonly used for vat photopolymerization processes. In this study, the layer thickness was set to 50㎛ referring to previous studies [3, 4] .

Comment: Insert he reference that have determined the parameters applied in the topic 2.4

Answer: Thank you for your good suggestion. Added references to determined parameters on lines 163-166 as you commented. The added parts was emphasized with red.

2.4. Biocompatibility test

2.4.1. In vitro cell culture and cell line

Cell viability and cytotoxicity assays were performed to evaluate the biocompatibility of molecules released from 3D-printed specimens according to the washing time. 3D-printed specimens were manufactured according to the manufacturer's instructions for the resin suggested in ISO 7405 and Research by Ian Gibson , I.G., and were manufactured with a diameter of 9 mm and a thickness of 2 mm to fit the size of a 48 well. Primary human gingival fibroblasts (HGFs; PCS-201-018, ATCC, Manassas, VA, USA) were used with Dulbecco’s modified Eagle’s medium (WelGene, Daegu, Korea) containing 10% fetal bovine serum (Thermo Scientific, Waltham, MA, USA), penicillin/streptomycin (100X, WelGene), and MEM nonessential amino acid solution (100X, WelGene).

Comment: What is the interaction between flexural strength and flexural modulus? And what means increase the values of one of these properties while the other is still the same? Discuss it and how it can affect your materials behavior in the clinical applicability.

Answer: Thank you for your kind review of the content. We defined the relationship between flexural strength and flexural modulus and what the experimental results mean were added to 496-503 and highlighted in red. Also we have added supplementary explanation to clinical applicability this study on line 509-522 of discussion and highlighted in red.

4. Discussion

As a result, it appears that the flexural modulus decreased with increasing washing time, but the flexural strength might not have. As flexural strength increases, the modulus of resilience increases, and as flexural modulus increases, the modulus of resilience decreases. Therefore, the modulus of resilience directly depends on the interaction between flexural strength and modulus. As shown in our experimental results, a decrease in flexural modulus without a change in flexural strength means an increase in modulus of resilience. The modulus of resilience is the amount of strain energy per unit volume that a material can absorb without permanent deformation. An increase in this value means an increase in the elasticity of the material.

The ISO 4049 standard for resin composites specifies only the requirement of the minimum FS value for the mechanical properties. But the elastic modulus of resin composites can have more clinical relevance since it provides resistance of material to deformation, which is correlated with the clinical performance [5]. Flexural modulus is a useful measure for determining whether a material that will break or rupture due to applied stress. A lower flexural modulus lowers the bending resistance and can cause problems such as repeated elastic deformation of a resilient restoration’s margin might lead to microleakage [6]. Furthermore a low flexural modulus could lead to restorative deformation under load, resulting in accelerated wear and hoop stress that may cause crown debonding [7, 8] and therefore it is considered that washing for a long time should be avoided. However, a material with high strength and low flexural modulus is suitable for dental trauma management because a flexible splint assists in healing [9]. Therefore, it seems that the washing time should be considered according to the purpose of use of the dental prosthetics.

Comment: Improve your discussion with regard to the saving time during the printed object processing.

Answer: Thank you for your kind review of the content. The added parts was emphasized with red.

4. Discussion

Therefore, our experimental results can be used as a basis for selecting a washing solution that suits specific situations and requirements. In addition, manufacturers suggest washing 3D-printed objects for longer than the recommended time if they are large or deep and narrow, but there are no exact guidelines for this. According to the results of the present study, washing for up to 90 min does not cause critical defects in mechanical properties, which can help when determining the optimal washing time according to the shape and size of specific objects. furthermore, as a result of the biocompatibility test of this study, cytotoxicity was greatly reduced when washing for more than 15 minutes, and most of the cell morphology was maintained normally. Also, the modulus decreased as the washing time increased, but in most groups, 15 minutes of washing did not cause any critical defects. Therefore, it is possible to reduce unnecessary washing time beyond that by recommending a minimum washing time of 15 minutes in object processing.

Reference

  1. Standards, I., ISO 7405: Preclinical evaluation of biocompatibility of medical devices used in dentistry-Tests methods for dental materials. 1997, International Organization for Standardization Geneva. Switzerland.
  2. Ian Gibson, I.G., Additive manufacturing technologies 3D printing, rapid prototyping, and direct digital manufacturing. 2015, Springer.
  3. Voet, V.S., et al., Biobased acrylate photocurable resin formulation for stereolithography 3D printing. ACS omega, 2018. 3(2): p. 1403-1408.
  4. Chung, Y.-J., et al., 3D printing of resin material for denture artificial teeth: chipping and indirect tensile fracture resistance. Materials, 2018. 11(10): p. 1798.
  5. Jun, S.-K., et al., Investigation of the correlation between the different mechanical properties of resin composites. Dental materials journal, 2013. 32(1): p. 48-57.
  6. Awada, A. and D. Nathanson, Mechanical properties of resin-ceramic CAD/CAM restorative materials. The Journal of prosthetic dentistry, 2015. 114(4): p. 587-593.
  7. Lawson, N.C., R. Bansal, and J.O. Burgess, Wear, strength, modulus and hardness of CAD/CAM restorative materials. Dental Materials, 2016. 32(11): p. e275-e283.
  8. Ilie, N. and R. Hickel, Investigations on mechanical behaviour of dental composites. Clinical oral investigations, 2009. 13(4): p. 427-438.
  9. Shirako, T., et al., Evaluation of the flexural properties of a new temporary splint material for use in dental trauma splints. Journal of dental sciences, 2017. 12(3): p. 308-310.

Reviewer 2 Report

Authors have performed nice studies on a few 3D-printed dental resin materials. The topic is interesting and good experiments are chosen. However, I have a few concerns to be answered before suggesting the work for publication. - The main problem I see rises from the fact that, except for the biocompatibility results, almost all tests show similar results for various conditions. It means various situations almost did not affect the 3D-printed materials. I suggest adding one or two other materials/conditions as control(s). Which materials are affected? Which conditions affect the printed materials? Please add them to the figures. Flat curves in most of the plots do not provide sufficient data for the reader(s). - Please provide figures of the printed parts. - Can authors also include the effect of some (acid and base) foods, saliva, etc. on the printed parts? I will be happy to re-read the manuscript if the authors choose to answer the provided comments.

Author Response

Report 2

Authors have performed nice studies on a few 3D-printed dental resin materials. The topic is interesting and good experiments are chosen. However, I have a few concerns to be answered before suggesting the work for publication.

Comment: The main problem I see rises from the fact that, except for the biocompatibility results, almost all tests show similar results for various conditions. It means various situations almost did not affect the 3D-printed materials. I suggest adding one or two other materials/conditions as control(s). Which materials are affected? Which conditions affect the printed materials? Please add them to the figures. Flat curves in most of the plots do not provide sufficient data for the reader(s).

Answer: Thank you for your kind review and good suggestions. I think adding another condition's material and condition as a control is a good suggestion. What you said is also an important factor, what we focused on in this study was the effect of different washing times and washing solutions on the materials during manufacturing. Also, there are some concerns about adding values by testing only the control group. First, although 3D printing technology has advanced, predictive capabilities at present are not accurate enough to fully understand and compensate for variations in shrinkage and residual stresses that are scan pattern or geometry dependent [1]. Second, biocompatibility testing using cells also has many differences depending on the experimental period. Because the cell is very sensitive, the cell's properties can change under the influence of many factors such as the passage of the cell, the culture environment, and the operator's technique. Therefore, I think that it may not be appropriate to compare the experimental group that has already been conducted with the control group that has been tested under different conditions. The two materials selected in our study are among the most used resins in clinical practice. What you mentioned is also an important factor, so it would be better if further follow-up studies could be done on what you mentioned. Therefore, the need for additional research was added in lines 539-541 and highlighted in red.

I think it is very important to point out that the mechanical strength test results showed similar results.  Although this study showed similar results in mechanical strength according to the washing time, to prevent the variation which can caused by different of materials, washing solutions and washing times, we have designed our experiments referring to previous studies which followed ISO standards.

 As a consequence, our results showed trends that consistent with the results of previous studies[2] as we compared in discussion Although there are insufficient studies on the effect of washing time on flexural strength, it is believed that this study can be the basis for other studies in the future. Various conditions affecting mechanical strength have been edited with supplemental references to the content in lines 536-539 and highlighted in red.

4. Discussion

In addition, the flexural strength and surface defects of 3D-printed resin seem to be more affected by aging and the printing direction [3-5], surface treatment [6, 7], printing method [8], and postcuring temperature and time [9, 10] than by any changes to the washing time within the upper limit of 90 min. Also, in order to more clearly confirm the effect of washing time on mechanical properties, it is considered that a follow-up study with a control group of other materials or conditions is needed.

Also, in our study, the 0min group was set as the control group and the experiment was conducted. Accordingly, the graph and name have been modified to make it easier for readers to understand. I added something for the Control group on line 138-139 and marked all modifications in red.

2.2. Washing the 3D-printed specimens and the postcuring process

Before washing, the support structures and residual resin of each 3D-printed specimen were removed, and then it was placed in a 3D printing washer (Twin Tornado, MEDIFIVE, Incheon, Korea) and washed with IPA or TPM solution for 3, 5, 10, 15, 30, 60, and 90 min to remove the residual resin monomer from the surface (Table 2). A nonwashing group was used as a control group (CTR).

Comment: Please provide figures of the printed parts.

Answer: Thank you for your good comment. Added Figure 1 was emphasized with red.

Comment: Can authors also include the effect of some (acid and base) foods, saliva, etc. on the printed parts?

Answer: Thank you for your kind review of the content. In this study, limitation was that the biocompatibility and mechanical testing did not reproduce the oral environment because in vitro test. We have edited supplementary explanation for better understanding on line 637-639 of suggesting that limitation of this study. The corrected parts was emphasized with red.

4. Discussion

One limitation of this study was the use of a single type of washer and washing method. Different types of ultrasonic equipment or solutions with temperature control of the solvent may produce different results. Another limitation was that the biocompatibility and mechanical testing did not reproduce the oral environment include the effect of acid and base foods, saliva, etc. Moreover, the accuracy of a 3D-printed part finally produced by focusing only on mechanical and biological properties may vary depending on various parameters.

Reference

  1. Ian Gibson, I.G., Additive manufacturing technologies 3D printing, rapid prototyping, and direct digital manufacturing. 2015, Springer.
  2. Xu, Y., et al., Effect of post-rinsing time on the mechanical strength and cytotoxicity of a 3D printed orthodontic splint material. Dental Materials, 2021. 37(5): p. e314-e327.
  3. KEßLER, A., R. Hickel, and N. Ilie, In vitro investigation of the influence of printing direction on the flexural strength, flexural modulus and fractographic analysis of 3D-printed temporary materials. Dental Materials Journal, 2021. 40(3): p. 641-649.
  4. Väyrynen, V.O., J. Tanner, and P.K. Vallittu, The anisotropicity of the flexural properties of an occlusal device material processed by stereolithography. The Journal of prosthetic dentistry, 2016. 116(5): p. 811-817.
  5. Shim, J.S., et al., Printing accuracy, mechanical properties, surface characteristics, and microbial adhesion of 3D-printed resins with various printing orientations. The Journal of prosthetic dentistry, 2020. 124(4): p. 468-475.
  6. Asli, H.N., et al., Effect of different surface treatments on surface roughness and flexural strength of repaired 3D-printed denture base: An in vitro study. The Journal of Prosthetic Dentistry, 2021.
  7. Thunyakitpisal, N., P. Thunyakitpisal, and C. Wiwatwarapan, The effect of chemical surface treatments on the flexural strength of repaired acrylic denture base resin. Journal of Prosthodontics: Implant, Esthetic and Reconstructive Dentistry, 2011. 20(3): p. 195-199.
  8. Park, S.-M., et al., Flexural Strength of 3D-Printing Resin Materials for Provisional Fixed Dental Prostheses. Materials, 2020. 13(18): p. 3970.
  9. Bayarsaikhan, E., et al., Effects of Postcuring Temperature on the Mechanical Properties and Biocompatibility of Three-Dimensional Printed Dental Resin Material. Polymers, 2021. 13(8): p. 1180.
  10. Li, P., et al., Postpolymerization of a 3D-printed denture base polymer: Impact of post-curing methods on surface characteristics, flexural strength, and cytotoxicity. Journal of Dentistry, 2021: p. 103856.

Round 2

Reviewer 2 Report

- The authors have not answered the comments appropriately. As I mentioned in the last review report, as the MAIN problem with the work, they need to include some test results to make the charts valuable. Now, the manuscript is full of flat curves, from which not much data can be gained. The authors have actually skipped answering this important problem. 

- Authors need to add the real figures of the printed parts. Just providing a schematic does not help.

- Authors also have skipped the suggestion of including the effect of food, which is super important for this study.

All in all, authors have not answered the comments appropriately and  they've skipped almost all  of them. I do NOT suggest publishing this work in this form. Major revision is needed.